# Mitogen-Activated Protein Kinases (MAPKs) and Enteric Bacterial Pathogens: A Complex Interplay

**DOI:** 10.3390/ijms241511905

**Published:** 2023-07-25

**Authors:** Ipsita Nandi, Benjamin Aroeti

**Affiliations:** Department of Biological Chemistry, Alexander Silberman Institute of Life Sciences, The Hebrew University of Jerusalem, Jerusalem 9190410, Israel; ipsita.nandi@mail.huji.ac.il

**Keywords:** MAP kinases, enteric bacterial pathogens, diarrheal diseases, type III secreted effectors, cholera toxins, host-pathogen interactions, inflammatory responses, MAPK inhibitors, anti-microbial treatments

## Abstract

Diverse extracellular and intracellular cues activate mammalian mitogen-activated protein kinases (MAPKs). Canonically, the activation starts at cell surface receptors and continues via intracellular MAPK components, acting in the host cell nucleus as activators of transcriptional programs to regulate various cellular activities, including proinflammatory responses against bacterial pathogens. For instance, binding host pattern recognition receptors (PRRs) on the surface of intestinal epithelial cells to bacterial pathogen external components trigger the MAPK/NF-κB signaling cascade, eliciting cytokine production. This results in an innate immune response that can eliminate the bacterial pathogen. However, enteric bacterial pathogens evolved sophisticated mechanisms that interfere with such a response by delivering virulent proteins, termed effectors, and toxins into the host cells. These proteins act in numerous ways to inactivate or activate critical components of the MAPK signaling cascades and innate immunity. The consequence of such activities could lead to successful bacterial colonization, dissemination, and pathogenicity. This article will review enteric bacterial pathogens’ strategies to modulate MAPKs and host responses. It will also discuss findings attempting to develop anti-microbial treatments by targeting MAPKs.

## 1. Introduction

Infectious pathogens are significant causes of diarrheagenic diseases, responsible for the annual deaths of approximately 0.8 million people, mainly children, worldwide [1,2]. Upon ingestion, the bacterial pathogens *Escherichia (E) coli*, *Shigella*, *Salmonella*, *Yersinia*, *Cholera*, and *Listeria* can cause acute infectious diarrhea (gastroenteritis) in humans [3,4]. Despite intensive research, the detailed molecular mechanisms underlying their mode of action still need to be deeply understood. Understanding these mechanisms is vital for developing novel anti-microbial treatments because pathogenic bacteria often produce antibiotic resistance [5,6]. Interestingly, a common characteristic these pathogens share is the ability to intercept the signaling cascades of the evolutionarily conserved NF-ĸB and host mitogen-activated protein kinases (MAPKs) [7,8,9,10,11], making the MAPKs–bacterial pathogen interface attractive targets for potential therapeutics [12,13,14]. 

The MAPK signaling pathways consist of a cascade of protein kinases activated sequentially: three canonical [extracellular signal-regulated kinase (Erk)1/2, c-Jun NH2-terminal kinase (JNK), p38 MAPK; Erk5 Figure 1A–D) and two non-canonical (Erk3/4 and Erk7/8) [15,16,17]. The activation of MAPK cascades begins with extracellular stimuli (cytokines, growth factors, stress factors, etc.). These signals are transmitted intracellularly by activating a set of cytoplasmic proteins termed ‘activators,’ which consist of small GTP binding proteins (e.g., Ras and Rho GTPases) and other host proteins. Then, signals are transmitted downstream by MAPKs organized in three-to-five interlinked tiers. In each tier, members of the MAPK family activate each other, typically by adding phosphate groups to serine, threonine, or tyrosine amino acids present in the Ser-Xaa-Ala-Xaa-Ser/Thr or Thr-Xaa-Tyr consensus sequences of activation loops. The activators first activate MAPKKK (MAP kinase kinase kinase) by several mechanisms, including phosphorylation, interaction with small GTPases, proteolysis, ubiquitination, and binding to regulatory or scaffold proteins [18]. Then, the MAPKKK activates MAPKK (MAP kinase kinase) by specific phosphorylation. The MAPKK subsequently activates (phosphorylates) the MAPKs. The activated MAPKs can then phosphorylate various substrate proteins, including transcription factors [e.g., activator protein 1 (AP-1), activating transcription factor 2 (ATF2), myocyte enhancer factor 2 (MEF2), c-Jun, and more], that coordinate the expression of downstream target genes controlling cellular activities [16,19,20,21]. The cellular responses depend on the activated MAPK pathway, e.g., the Erk1/2 pathway is associated with cell proliferation and survival. In contrast, the c-Jun N-terminal kinase (JNK) and p38 pathways are often implicated in stress responses, inflammation, and apoptosis. The Erk5 MAPK pathway responds to both growth signals and specific stresses.

In the onset of infection, evolutionarily conserved structural elements residing on the bacteria’s exterior, termed PAMPs, e.g., flagellin, peptidoglycan (PGN), and lipopolysaccharides (LPS), are recognized by host PRRs, e.g., TLRs, to trigger two major signaling pathways: the Myd88/IRAK1,4/TRAF6/TAK1/IKK/NF-ĸB pathway and the Myd88/IRAK1,4/TRAF6/ TAB2,3)/TAK1/MAPK (JNK/p38)/ pathway (Figure 1E). These pathways lead to the activation of proinflammatory responses (e.g., the one involving AP-1 transcription factor that results in cytokine interleukin-8 (IL-8) production [22]) involved in innate immunity [18,23]. Notably, some components of the MAPK pathways (e.g., c-Raf, MEKKs, and TAK1) can also trigger the NF-kB pathway [24]. 

In addition to stimulating innate immunity, bacterial pathogens developed sophisticated mechanisms to suppress or activate the MAPK pathways by targeting specific components of the pathways. A common strategy involves a multi-protein virulence-associated nanomachinery with a needle-like structure called the type III secretion system (T3SS) [25]. T3SS injects bacterial proteins, termed effector proteins, from the bacterial cytoplasm into the infected cells. These effectors are encoded by specific genes located in the locus of enterocyte effacement (LEE) and non-LEE pathogenicity islands of the bacterial chromosome. Upon translocation, the effectors act as networks that hijack and subvert diverse eukaryotic cell processes, determining the quality of cell infection, tissue tropism, and more [26,27,28]. Another strategy involves the secretion of toxic bacterial proteins, termed toxins. These bacterial-secreted components (effectors and toxins) interact and modulate the activity of various host cell proteins, including MAPKs [18,29], thereby contributing to bacterial pathogenicity. The impacts of these toxic proteins on individual MAPKs are schematically depicted in Figure 2 and summarized in Table 1.

In this article, we systematically review data regarding the modes by which extensively studied enteric bacterial pathogens hijack and modulate the activity of host MAPKs and their impact on themselves and the host. Furthermore, we review advances in discovering small molecular weight molecules that inhibit the activity of MAPKs and bacterial infection as potential platforms for future therapeutics.

## 2. Bacteria, Toxins, Effectors and Modes of MAPK Subversion

### 2.1. Enteropathogenic and Enterohemorrhagic Escherichia coli

Enteropathogenic *Escherichia coli* (EPEC) and the Shiga toxin-producing enterohemorrhagic *E. coli* (EHEC) are well-studied Gram-negative extracellular diarrheagenic bacterial pathogens that colonize the human small and large intestines, respectively [30,31]. While EPEC mainly causes acute pediatric diarrhea, EHEC induces diarrhea complicated by hemorrhagic colitis and hemolytic uremic syndrome in adults [32,33]. *Citrobacter (C.) rodentium* is a natural extracellular enteric pathogen that causes acute gastroenteritis in mice. EPEC, EHEC, and *C. rodentium* are members of the ‘attaching and effacing’ (A/E) family of gastrointestinal bacterial pathogens. A/E pathogens induce localized lesions in the intestinal tissue, characterized by the firm attachment of the bacteria to the apical plasma membrane of the intestinal epithelial cells and the effacement of juxtaposed microvilli. In addition, the binding of A/E pathogens to the host epithelial cells forms a filamentous (F)-actin-rich pedestal-like structure, on top of which the bacterium resides. EPEC and EHEC infection induces IL-8 production via PAMP-TLR5-mediated activation of MAPK and the NF-κB pathways [34,35]. Type III-secreted effector proteins translocated by these pathogens into the host cells either stimulate or inhibit these pathways, as described below. 

**Tir** (translocated intimin receptor) is the first translocated type III secreted effector, which constructs the F-actin-rich pedestal by interacting with the bacterial outer membrane protein (OMP) intimin [36,37]. However, Tir has also been shown to play another vital role during infection, which is the downregulation of innate immune responses of the host via a sequence harbored by its cytoplasmic domain that is similar to immunoreceptor tyrosine-based inhibition motifs (ITIMs) [38]. ITIMs are conserved amino acid sequences found in the cytoplasmic domains of many tyrosine-phosphorylated receptor families expressed in immune cells [39] whose role is to downregulate immune responses [40]. Upon translocation into the host cells, specific cytoplasmic tyrosine residues within the ITIM motif of Tir (Tyr483 and Tyr511) undergo phosphorylation, leading to the recruitment of the host cellular tyrosine phosphatase, src homology region 2 domain-containing phosphatase-1 or 2 (SHP-1 or SHP-2) proteins, which subsequently inhibit the TRAF6 autoubiquitination (needed for its activation as a ubiquitin ligase [41]), and the downstream NF-ĸB/MAPK signaling pathways [38,42]. Notably, the Tir-mediated inhibition of innate immune responses to infection was also shown in in vivo experiments using the *C. rodentium* infection model [38].

The actin-modulating effector, **EspH**, has been suggested to act as an anti-GEF to suppress host Rho GTPases (Cdc42 and Rac1) and induce host cell cytotoxicity [43,44,45,46]. Intriguingly, EspH has also been shown to suppress host Erk by its spatial segregation from tetraspanin CD81 microdomains, and a C-terminal 38 amino acid domain of the effector protein is essential for mediating the process. Moreover, EspH selectively inhibited the tumor necrosis factor-α (TNF-α)-induced Erk signaling pathway [47]. Recent mechanistic studies have shown that the ability of EspH to inhibit Rho GTPases (primarily Rac1) occurs by binding the GTPase activating protein (GAP) domain of the host active Bcr related (Abr) protein [48], thereby stimulating the Rho GAP activity and host cell death. These studies also suggested that the C-terminal 38 amino acid segment of EspH is involved in Abr binding, suggesting that the EspH-dependent inhibition of Rho GTPases and Erk are linked processes. Since Erk is activated by Rac1 and Cdc42 signaling [49], EspH may inhibit Erk by exerting Abr-dependent Rac1 inhibition. 

In contrast to EspH, the mitochondrion-associated protein effector (**Map**) of EPEC activates Cdc42 through a Trp-xxx-Glu (WxxxE) Rho-guanine exchange factors (Rho-GEFs) motif [50], and the epidermal growth factor receptor (EGFR)/MAPK (Erk, MEK, p38) signaling pathways [51,52]. The activation of MEK and Erk was WxxxE-independent. The activation of these MAPKs was, however, dependent on a MAP mitochondrial cytotoxicity motif through which the effector causes mitochondrial membrane potential disruption, Ca^+2^ efflux from mitochondria to the cytoplasm, and the stimulation of sheddase activity of a disintegrin and metalloproteinase domain-containing protein 10 (ADAM10). ADAM10 activation has been reasoned to cause the release of EFG/Betacellulin, which activates (phosphorylates) the EGFR-MEK-Erk signaling pathway, leading to host cell apoptosis [52]. 

In EPEC-infected intestinal epithelial cells and infected murine models, the effector non-LEE-encoded effector H1 (**NleH1)** suppresses the activation of Erk1/2 and p38 [53], and the closely related effector **NleH2** suppresses p38 only. The NleH effectors possess a kinase domain that manipulates the NF-ĸB pathway [54]. However, the NleH-mediated MAPK suppression is not dependent on kinase activity. Since MAPK activation is implicated in the progression of inflammatory bowel diseases, e.g., colitis (further discussed below and see Ref. [55]), it has been hypothesized that NleH1-mediated inhibition of MAPKs confers protection against recovery from colitis [53]. 

In vitro and in vivo studies have shown that the effector **NleC** is a zinc metalloprotease that disrupts the NF-ĸB activation pathway by cleaving the p65, one of the five components that form the NF-κB transcription factor family [56,57,58]. In addition, NleC has also been identified as an effector suppressing IL-8 release by inhibiting both NF-ĸB and the p38 MAPK activation [59]. However, the mechanism by which NleC inactivates p38 remains unknown. 

**NleD** is another zinc metalloprotease effector protein that suppresses the MAPK signaling by directly cleaving the c-JNK and p38 MAPKs, but not Erk [60,61]. NleD cleaves the indicated MAPKs within the activation loop that links the MAPK N-and C-termini. The loop harbors a conserved threonine and tyrosine separated by glycine or proline, i.e., a threonine-X-tyrosine (TXY) motif. MAPK activation is achieved by phosphorylating the motif’s threonine and tyrosine residues. NleD cleaves the MAPKs between the Gly/Pro and the Tyr of the TXY motif rendering them permanently inactive. Notably, NleD and NleD-like proteins are expressed by the three A/E pathogens, as well as by *Salmonella enterica*, plants, and symbiotic bacteria, indicating that NleD represents a family of zinc metalloproteases that target MAPKs. Finally, the inhibitory effect on MAPKs by NleD may result in the inhibition of AP-1-dependent gene transcription and innate immune responses [62]. 

**NleE** is an EPEC effector that disrupts the NF-ĸB signaling by inhibiting the MAPKKK, TAK1 [63,64,65]. TAK1 binds to TAB2,3. The assembly of TABs is promoted by E3 ubiquitin ligases, which catalyze the synthesis of K63-linked polyubiquitin chains that preferentially bind to TAB2 and TAB3 subunits, resulting in the assembly and activation of the TAK1–TAB2/3 complex [66]. NleE is a specific cysteine methyltransferase that methylates a conserved cysteine residue in the Npl4-like Zinc Finger domains of TAB2/3. This modification abrogates the TAB2/3 ability to bind ubiquitin chains, thereby preventing the activation of TAK1 and downstream NF-ĸB [63]. As TAK1 signals to both MAPKs and NF-ĸB (Figure 1), it is possible that the inhibition of TAK1 by NleE also leads to the inhibition of MAPKs. However, this hypothesis awaits further exploration. Nevertheless, NleE inhibits the production of proinflammatory chemokines, IL-6, IL-8, and TNF [64,67,68].

In summary, A/E pathogens indirectly inhibit MAPK signaling pathways, e.g., by the Tir and EspH effectors, which target TRAF6 and Rho GTPases, respectively, and directly, e.g., by NleD, which exerts metalloprotease activity on MAPKs. 

### 2.2. Shigella

The *Shigella* (S.) Enterobacteriaceae family consists of four species: *S. dysenteries*, *S. boydii*, *S. flexeneri,* and *S. Sonnei*. *S. flexneri* are invasive Gram-negative bacteria and the etiological agents of endemic Shigellosis (bacterial dysentery), responsible for about a third of the annual deaths caused by infectious enteric diseases [69,70]. These bacterial pathogens invade the intestinal barrier by transcytosis through the intestinal microfold (M) cells. They use T3SS to colonize and kill macrophages in the submucosa. Following the induction of cell death, bacteria are released from the dying cells and invade the intestinal epithelium by infecting their basolateral surface. Once internalized into the cells, *S. flexneri* reaches the host cytoplasm and uses a unique actin-based motility mechanism to move and spread to adjacent cells [71]. 

Studies on infected macrophages suggested that the OMPs of *S. flexneri* stimulate the host TLR2 and TLR6, followed by MyD88/TRAF6, NF-ĸB, and the p38 MAP kinase signaling pathways [72]. In addition, studies on infected intestinal epithelial cells suggested a role for the *S. flexneri* LPS-mediated activation of Erk in bacterial invasion through the basolateral surface of these cells [73]. The *Shigella* PGN is recognized by nucleotide oligomerization domain (NOD)-like receptors NOD1 and NOD2, which control both MAPK and NF-κB proinflammatory signaling (reviewed in [74]). *S. flexneri* induces a massive secretion of the proinflammatory IL-8 in intestinal epithelial cells. Interestingly, the proinflammatory responses involved the activation of NF-ĸB and the MAPKs c-JNK, Erk, and p38 signaling pathways, which rapidly propagated from infected to uninfected cells via gap junctions, thereby stimulating proinflammatory responses in uninfected cells [75]. 

**Outer *Shigella* protein F (OspF)** is a T3SS *S. flexneri* effector with a phosphothreonine lyase activity. It contains a motif mimicking the canonical D motif of many MAPK substrates required for MAPK docking [76]. Once the MAPKs are docked on OspF, OspF exerts its phosphothreonine lyase activity, catalyzing an irreversible elimination reaction that converts phosphothreonine into dehydrobutyrine [77]. This activity removes the phosphate moiety from a phosphothreonine residue in the TXY motif in the activation loop of the MAPKs (c-JNK, p38, and Erk 1/2), thereby inactivating them in mammalian [76,78,79] and yeast [80] cells. In mammalian cells, the inactivation of MAPKs results in the downregulation of genes involved in the inflammatory and immune responses (e.g., c-Fos, IL-8) [79]. In a guinea pig model of Shigellosis, a phosphothreonine lyase activity was carried out by OspF to inactivate Erk and the mitogen- and stress-activated protein kinase 1 (MSK1) proteins needed for the phosphorylation and activation of the heterochromatin protein 1 (HP1γ) transcriptional regulator. The impact of this epigenetic activity is complex, as it results in the inhibition or stimulation of transcription responses (e.g., of IL-8, CD44, NF-ĸβ) involved in immune defenses and the repair of the infected mucosa [81]. 

At the onset of infection, **OspB**, another type III secreted effector of *S. flexneri*, activates Erk1/2 and p38 MAPKs. Activating these MAPK cascades promotes the production and secretion of the polymorphonuclear (PMN) leucocytes chemoattractant metabolites required for eliciting a complete inflammatory response in vitro (Caco-2 and HeLa cells) and in vivo (Guinea pig) infection models [82]. It has been postulated that at an early *Shigella* infection phase, the OspB injected into the enterocytes’ cytosol stimulates the Erk1/2/p38 signaling. The activation of the MAPKs, in turn, activates the cytosolic phospholipase A2 enzyme, releasing arachidonic acid from the endomembrane. The arachidonic acid is metabolized into mediators belonging to the eicosanoid class of lipids, which serve as chemoattractants for PMN leucocytes. These chemoattractants are released from the apical surface of the epithelial cells, forming a chemotactic gradient that directs PMN migration across the epithelium at the site of infection. It has been reasoned that PMN migration from the basolateral to the apical milieu disrupts the tight junction intestinal barrier, thereby maximizing the host cell surface susceptibility and the capacity for bacterial infection. At a later infection stage, injected OspF inhibits the MAPKs and the innate immune responses [82]. 

In the yeast *Saccharomyces cerevisiae* model, expressed *Shigella* E3 ubiquitin-protein ligase **IpaH9.8** effector has been shown to interrupt the pheromone response signaling by promoting the proteasome-dependent degradation of the MAPKK Ste7. Further in vitro studies have demonstrated that IpaH9.8 displays an E3 ubiquitin ligase activity towards the Ste7 [83,84]. Finally, studies in mammalian cells and mice have shown that IpaH9.8 also plays a vital role in modulating the host inflammatory responses by ubiquitinating and sending for proteasomal degradation the NF-ĸB essential modulator (NEMO) [85] and the human interferon-inducible guanylate-binding proteins [86,87]. This enhance bacterial actin-dependent motility and the suppression of anti-bacterial defense activities. 

In summary, *Shigella* effectors (e.g., OspB) can activate or inactivate (e.g., OspF and Ipa9.8) MAPKs. OspF and IpaH9.8 irreversibly inactivate host MAPKs by changing their biochemical properties, e.g., by phosphothreonine lyase activity, or by eliminating them, e.g., by ubiquitination and proteosomal degradation. 

### 2.3. Yersinia

The Gram-negative facultative anaerobic bacterial pathogen *Yersinia pestis* is the causative agent of plague, one of the deadliest diseases in human history [88]. Two related human pathogens, *Yersinia pseudotuberculosis* and *Yersinia enterocolitica*, cause gastroenteritis [89]. *Yersinia* utilizes the bacterial adhesins Invasin A (InvA) and *Yersinia* adhesin A (YadA) to invade mammalian host cells. InvA binds to the β1-integrin receptors of the host cell surface, facilitating bacterial invasion and transcytosis through the intestinal epithelial barrier. Interestingly, it has been reported that InvA-β1-integrin receptors interactions activate a signal cascade involving Rac1/MAPK/NF-kB, facilitating proinflammatory cytokines [90]. 

The virulence of *Yersinia* in macrophages depends on the T3SS and injected effector proteins called *Yersinia* outer proteins (Yops) [91]. One of these effectors, **YopJ** of *Y. pestis*, a homolog of **YopP** of *Y. enterocolitica,* plays an essential role in *Yersinia*-induced cell death programs (apoptosis) by deactivating the MAPK and NF-κB signaling pathways [92,93,94,95]. 

Landmark studies discovered that YopJ is an acetyltransferase, using acetyl-coenzyme A to modify critical serine, threonine, and lysine residues in the activation loop of MAPKKs, including MEK2, MKK4, MKK6, MKK7, and the upstream MAPKKK-TAK1 in mammalian cells and *Drosophila* [96,97,98]. Upon acetylation, these MAPKs cannot undergo the phosphorylation needed for their activation and signaling. By inhibiting these signaling pathways, YopJ induces programmed cell death and the inhibition of proinflammatory signal transduction mediated through the protective cytokines TNFα and IL-8. The mechanism by which YopJ inhibits MAPK signaling is evolutionarily conserved because YopJ also inhibits the MAPK pathways in the yeast *Saccharomyces cerevisiae* by acetylating the yeast MKK, Pbs2 [99,100]. 

**YopE** and **YopT** are *Yersinia* type III secreted effectors which downregulate Rho family GTPases by different mechanisms. **YopE** inactivates the Rho GTPases through its GAP activity [101,102,103,104]. **YopT** is a cysteine protease that proteolytically cleaves the lipid modification of the Rho GTPases, resulting in their release from membranes and deactivation [104,105]. Studies of infected cell cultures have shown that the YopE and YopT expression inhibited the activation of the downstream c-JNK, Erk, and NF-ĸB, thereby preventing IL-8 production. However, comparison studies in mammalian cells and an infected mouse model showed that YopT only moderately affected these responses [106]. 

In summary, *Yersinia* effectors indirectly inhibit MAPKs, e.g., by YopE, which inactivates RhoGTPases, and directly by exerting enzymatic activities. YopJ exerts acetyltransferase activity, and YopT, a cysteine protease, executes proteolytic activity on the MAPKs. 

### 2.4. Salmonella

*Salmonella enterica* serovar Typhimurium causes gastroenteritis and systemic infections. It is one of the leading causes of food-borne illnesses in the industrial world [107]. This pathogen colonizes multiple niches in the host, alternating between extracellular and intracellular lifestyles. This bacterium utilizes two distinct T3SSs encoded by the *Salmonella* pathogenicity island-1 (SPI-1) and -2 (SPI-2), injecting into the host cells an array of effectors [108,109]. Extracellular bacteria mainly use SPI-1 effectors to promote bacterial invasion, and the SPI-2 effectors promote the bacterial replication and modulation of host immune signaling [109,110,111,112]. While external *Salmonella* components act as PAMPs that activate MAPKs and anti-microbial immune responses [113,114], translocated effectors target MAPKs to elicit pro- or anti-inflammatory activities [109,115,116,117]. 

The SPI-1 *Salmonella* outer proteins **SopE/SopE2** and **SopB** effectors have redundant functionalities. SopE/SopE2 harbors a WxxxE Rho GEF motif, which mainly directly activates Rac1. SopB indirectly activates host RhoGTPases by activating the SH3-containing GEF of RhoG [118]. In addition, these Rho GEF effectors stimulate the downstream MAPKs, Erk1/2, p38, JNK, and NF-ĸB to promote F-actin-based plasma membrane ruffles needed for the bacterial internalization and proinflammatory (IL-8) immune responses [119].

The *Salmonella* translocator effector C, **SteC**, is an SPI-2 effector delivered into the host cells by intracellular bacteria, exerting kinase activity through its kinase domain. It phosphorylates and activates MKK, activating the MKK/Erk/myosin light chain kinase/Myosin IIB signaling pathway. In addition, SteC phosphorylates MEK1 Serine200, which is required for SteC-induced MKK activation. This kinase activity of SteC generates an F-actin cytoskeletal meshwork that surrounds the bacterium and contributes to the control of bacterial survival in cultured cells, but not in an infected mouse model [120,121]. 

The *Salmonella* protein tyrosine phosphatase **(SptP)** is an SPI-1 effector. SptP seems to reverse the *Salmonella* stimulatory effects during entry by downregulating the RhoGTPases Cdc42 and Rac through a GAP domain located at its N-terminus and by activating a tyrosine phosphatase activity exerted by its C-terminus to downregulate the activation of Erk1/2, c-JNK, and IL-8 production [122]. Subsequent studies suggested that the Erk1/2 inhibition is achieved by SptP-mediated Raf-1 inhibition. Both the tyrosine phosphatase and the GAP activities have been suggested to be involved in the inhibition of Raf-1/Erk. Moreover, deleting the *Sptp* gene enhanced the capacity of Salmonella to induce TNF-α secretion [123]. However, the precise molecular mechanisms underlying SptP inhibitory effects remain unknown. 

The SPI-1 **AvrA** (a close homolog of YopJ [124]) effector possesses acetyltransferase activity to acetylate a specific serine residue of MKK4 and MKK7. As a result, these MAPKKs can no longer be phosphorylated and activated, leading to the selective inhibition of the downstream c-JNK and NF-ĸB signaling pathways in mammalian cultured cells, yeast, Drosophila, and murine models, resulting in the inhibition of inflammatory responses and cell death [125,126]. Interestingly, although not targeting the Erk pathway, AvrA is phosphorylated on a conserved Ser14 by Erk, demonstrating the complexity of the interplay between the effector and host MAPKs [126]. Notably, in vivo studies have suggested that AvrA contributes to bacterial survival during murine infection [127]. 

The SPI-1 Invasion plasmid antigen J, (**IpaJ)** is a protein effector with cysteine protease activity present in *Salmonella* and *Shigella* species that has not been (so far) linked to the ability of the effector to modulate MAPKs. Nevertheless, studies suggested that injected IpaJ inhibits the host NF-κB and the MAPK signaling pathways. IpaJ can prevent the ubiquitination and degradation of IκBα in the NF-κB signaling pathway and inhibit the phosphorylation of MEK and Erk in the MAPK signaling pathway through deubiquitylation and the inhibition of Ras, thereby downregulating proinflammatory responses, cellular growth and differentiation, cell survival, and apoptosis [128]. However, the detailed mechanism by which IpaJ targets the Ras/MAPK pathway remains unknown. 

The *Salmonella* plasmid virulence C, **SpvC**, is an effector that can be secreted by either the SPI-1 or SPI-2 and is essential for *Salmonella* virulence in mice. Like OspF in *S. flexneri*, SpvC exhibits a phosphothreonine lyase activity towards p38 [76]. Furthermore, SpvC phosphothreonine lyase activity has also been shown to target Erk1/2 and c-JNK [129]. However, in a mice infection model, where SpvC is required for systemic infection [130], only pErk was dephosphorylated by the effector protein [131]. Moreover, the SpvC-mediated inhibition of MAPKs reduces the expression of proinflammatory cytokines and neutrophil infiltration [129,131]. Finally, data have shown that the lyase activity of SpvC inhibits pyroptotic cell death by inhibiting autophagy and innate immunity in an Erk-dependent fashion [132,133].

In summary, *Salmonella* exploits effector proteins to inhibit (e.g., IpaJ, SptP, AvrA, SpvC) or stimulate (e.g., SopE and SteC) MAPK cascades. The indirect effects are contributed by effectors that target small GTPases (e.g., Ras, Rac1, CDC42), and the direct effects are exerted by enzymatic activities, e.g., SpvC-mediated phosphothreonine lyase activity and AvrA-mediated acetyltransferase activity. 

### 2.5. Vibrio cholerae and Vibrio parahaemolyticus

*Vibrio* is a genus of Gram-negative, facultative anaerobes which encompass several species that inhabit aquatic environments, including marine, estuarine, and freshwater habitats. *Vibrio cholerae* and *Vibrio parahaemolyticus* are intensively studied members of this genus. 

*Vibrio cholerae* is a highly contagious bacterium responsible for cholera, an acute diarrheal infection caused by ingesting contaminated food or water, affecting millions in countries with poor sanitation [134]. The pathogen expresses several essential virulence factors, enabling its efficient colonization of the human intestine. During *Vibrio cholerae* infection, various extracellular components of the bacteria trigger proinflammatory responses by activating NF-κB and MAPK signaling. For instance, upon attaching to host enterocytes, flagellin interacts with TLR5, prompting the production of IL-8 by activating p38, c-JNK, and Erk1/2 [135]. Similarly, OmpU, another extracellular protein primarily functioning as a porin, activates proinflammatory responses by activating p38 and c-JNK MAPKs through the TLR2 receptor [136].

*Vibrio cholerae* consists of multiple serogroups, two of which are the O1 and O139 serogroups, causing cholera outbreaks due to their ability to produce cholera toxins (CTXs) [134]. CTXs, released by the type 2 secretion system of the microbe, have two subunits, A (CTXA) and B (CTXB). The A–B toxin binds through the B subunit to the ganglioside GM1 on the host cell surface, facilitating the entry of the A subunit into the host endocytic system. Following retrograde transport to the endoplasmic reticulum, CTXA enters the host cell cytoplasm, where ADP ribosylates G proteins and continuously activates adenylate cyclase. This activity leads to increased cyclic AMP levels and the extensive secretion of water into the extracellular environment, thereby contributing to diarrhea [137,138]. In addition, studies have shown that CTXB can act as an immunomodulator in macrophages. CTXB induces the upregulation of MAPK phosphatase-1 (MKP1), a negative regulator of macrophage inflammatory response, to inhibit LPS-activated proinflammatory responses (TNFα and IL-6) by c-JNK and p38 [139,140].

Specific strains of *Vibrio cholerae* can produce accessory MARTX (multifunctional-autoprocessing repeats-in-toxin) toxins released to the extracellular milieu by an atypical type 1 secretion system. These toxins inactivate the host Rho GTPases, further suppressing the downstream MAPK signaling pathways, IL-8 production, and intestinal inflammation [141,142]. 

**VopE**, a close homolog of the *Yersinia* YopE, is a type III secreted effector of *Vibrio cholerae*, shown to perturb innate immunity by targeting mitochondria and modulating host mitochondrial dynamics [143]. Interestingly, in a yeast model, wild-type VopE, and a VopE mutated in the mitochondrial targeting sequence can disrupt MAPK signaling [144]. 

***Vibrio parahaemolyticus*** is a well-studied species of *Vibrio* genus, which causes acute enteric diseases. It thrives in salty environments and is commonly found in seafood, particularly raw or undercooked shellfish. *V. parahaemolyticus* possesses various virulence factors, including adhesins, toxins, and two types of secretion systems, T3SS1 and T3SS2 [145,146]. Effectors delivered by these T3SSs have been shown to play a crucial role in *V. parahaemolyticus* pathogenicity by manipulating host cell MAPKs [147]. 

One such effector protein is the T3SS1 **VopQ**. VopQ is a channel-forming effector that targets the host vacuolar (V)-ATPase, resulting in lysosome deacidification and inhibiting lysosome–autophagosome fusion. In vitro studies have shown that the VopQ-mediated disruption of V-ATPase activates the inositol-requiring enzyme 1 (IRE1) branch of the unfolded protein response, resulting in the induction of Erk1/2 phosphorylation and signaling. Interestingly, another T3SS1 effector, **VopS**, antagonizes the VopQ-mediated activation of the Erk1/2 and JNK pathway by AMPylation, i.e., covalently attaching an adenosine monophosphate (AMP) molecule to a threonine residue in the switch one region of Rho GTPases [148,149]. 

**VopA** is a T3SS2 effector protein that shares significant sequence homology with the YopJ-like effectors of *Yersinia* and *Salmonella*. VopA is an acetyltransferase that effectively inhibits MAPKs (Erk1/2, p38, and c-JNK) in mammalian and yeast cells, ultimately suppressing host innate immune responses without affecting the NF-κB pathway [150,151]. **VopZ**, another T3SS2 effector, inhibits MAPKs and NF-κB by deactivating TAK1, enabling intestinal colonization and diarrhea induction [152].

In summary, *Cholerae* uses diverse means to hijack and control the activity of host MAPKs. Secreted effectors, e.g., VopE and VopQ, control MAPKs by targeting host cell organelles. Other effectors, e.g., VopS and VopA, exert enzymatic activities to modify MAPKs and their regulators. VopZ attacks TAK1, which is central to activating several MAPKs. 

## 3. *Listeria monocytogenes*

*Listeria monocytogenes* is a Gram-positive, facultative, intracellular, food-borne invasive bacterial pathogen that causes listeriosis, a systemic infection manifesting as bacteremia. These pathogenic bacteria can invade tissues and organ barriers, including the intestinal tissue, causing meningoencephalitis in immunocompromised individuals and the elderly, and fetal-placental infection in pregnant women [153]. *L. monocytogenes* have evolved remarkable strategies to invade epithelial cells and phagocytes. Binding and entry into non-professional phagocytes are induced by binding the bacterial surface adhesins, internalin A (InlA) and InlB, to receptors on the host cells. Once internalized into the epithelial cells, the bacterial pathogen escapes from the host vacuolar compartment, reaching the host cell cytoplasm, where it moves by an actin-based motility mechanism and proliferates. Listeriolysin O (LLO) is a *Listeria monocytogenes* secreted pore-forming toxin, which, together with two bacterial phospholipases C, mediates the escape of internalized bacteria from the vacuole to the host cytoplasm [154]. 

*L. monocytogenes* has been observed to activate MAPKs upon bacterial attachment to cultured epithelial cells [155]. The activation effect is contributed by the bacterial InlB interactions with the host cell c-Met and E-cadherin on plasma membrane lipid rafts [156], through the Ras-Erk1/2 MAPK pathway [157]. LLO activates the Raf-MEK MAPK pathway [158,159]. Regardless of the indicated mechanisms, studies suggested that *L. monocytogenes* can rapidly activate the p38 MAPK pathway in infected macrophages [160]. Subsequent studies have interestingly shown that bacterial products in the host cytoplasm, but not within the vacuole, activate NF-ĸB and the p38 MAPK [161]. Finally, the significance of the p38 MAPK activation pathway in *Listeria* infection has also been demonstrated in an infected mouse model [162]. 

## 4. Concluding Remarks

The human intestine has a large surface area [163], which is permanently ‘bombarded’ by various microbial pathogens, including bacteria. However, the cells of the intestinal tissue (e.g., the epithelial cells and underlying immune cells) evolved a remarkable range of defense strategies that prevent bacterial infections [164]. One strategy involves sensing external bacterial components (e.g., LPS, PGN, and more) by host receptors activating the NF-ĸB and MAPK signaling cascades. These cascades, activated early upon pathogenic bacterial contact with the host, launch massive and efficient innate immune responses that often eradicate infection [20,21,165]. However, the bacteria, which co-evolved with such responses, ‘fight back’ by activating secretion systems, releasing toxins and effector proteins that eliminate, or diminish the ability of host cells to activate the MAPKs (see Figure 2, Table 1 and Refs. [10,11,29,166]). It is possible that the extent of this effect determines whether an acute infectious diarrheal disease will emerge, or not. 

The bacterial effectors and toxins developed diverse and impressive means to inactivate the MAPKs, ranging from indirect effects, i.e., by inhibiting their upstream regulators (e.g., Rho GTPases), to a direct impact, e.g., by acting as enzymes whose catalytic activity (metalloprotease, methyltransferase, acetyltransferase, and phosphothreonine lyase, and more) irreversibly inactivates the MAPKs. The enzymatic inhibitory strategy is interesting, as it has likely evolved to rapidly and irreversibly suppress the MAPKs. These drastic effects are probably needed to eliminate the host immune response and enable successful bacterial infection. In this context, it is worth pointing out that many different effectors also evolved the capacity to inactivate TAK1-TABs, which is a central signalosome for the activation of several MAPK pathways, including the NF-kB pathway (Figure 1), and the inflammatory responses initiated by them [66]. Therefore, it would be reasonable to hypothesize that the inhibition of TAK1/TABs combined with the irreversible inhibition of MAPKs by an enzymatic activity evolved as a common mechanism to efficiently overpower the host cell and to establish successful infection. 

In some cases, effector proteins activate rather than inhibit the MAPKs. At first glance, this may look odd because the process could launch a MAPK-dependent immune response that has a deleterious effect on the bacterial pathogen. However, the pathogen may exploit other MAPK effects for its benefit, such as remodeling the host cell actin cytoskeleton through MAPK-dependent phosphorylation activity. For example, the *Salmonella* SopE, SopB, and SteC effectors stimulate the activity of MAPKs to remodel the host actin cytoskeleton needed to form a *Salmonella*-containing vacuole, within which bacteria can safely reproduce. In summary, the fact that some effectors block the MAPK cascade whilst others induce it points to the existence of tight balancing activities caused by them. Although the mechanisms of these counterbalancing effects have been revealed in recent years, our understanding of how they act on MAPKs to achieve a firm infection is a significant challenge for future investigations. 

Do MAPKs responses indeed play an essential role in eliminating pathogenic bacteria? To address this, bacterial infection parameters must be examined in response to intentional MAPK inhibition, e.g., by drugs. Unfortunately, however, a limited number of studies have described the use of this approach. For example, Baicalin (5,6-Dihydroxy-4-oxygen-2-phenyl [1]4H-1-benzopyran-7-β-D-glucopyranose acid), a Chinese medicinal herb isolated from the root of *Scutellaria baicalensis Georgi*, protected mice against *Salmonella* infection [167]. Additional studies have shown that the Baicalin protective effect is due to inhibiting reactive oxygen species production, autophagy, and the TLR4/MAPK/NF-κB signaling pathways [168]. Another study showed that blocking the MAPK and NF-κB signaling pathways did not allow the exertion of YopJ-dependent cell death in macrophages [94]. Notably, the murine p38 MAPK was first identified as a kinase activated in response to the bacterial endotoxin LPS [169]. Interestingly, inhibiting p38 or JNK by small molecular weight compounds in a murine model has significantly attenuated inflammatory responses after a systemic LPS challenge [[170] reviewed in [165]].

Inflammatory bowel diseases (IBDs), including ulcerative colitis and Crohn’s disease, are chronic disorders of the gastrointestinal tract with a rising incidence in the pediatric population [171]. Although the etiology of these diseases’ emergence is unclear, microbial pathogens, including pathogenic *E. coli* strains, and MAPKs, have been implicated in triggering them [172,173,174,175]. Bacterial infections and the microbiome have also been suggested to play a role in other severe human inflammatory diseases in humans, such as the autoimmune disease rheumatoid arthritis [176,177]. Interestingly, MAPK inhibitors have been applied in treating inflammatory diseases in animal models and humans and have been shown to emerge as a promising approach for combatting inflammatory diseases (reviewed in [165]). Unfortunately, however, in many cases, patients are not responsive or experience severe side effects to such therapies [174,178,179]. Therefore, a better understanding of the molecular mechanisms by which pathogenic bacteria interface the MAPK pathways may lead to the development of rational design drugs with better efficacy and less off-target effects to treat devastating inflammatory diseases in humans.

## Figures and Tables

**Figure 1 ijms-24-11905-f001:**
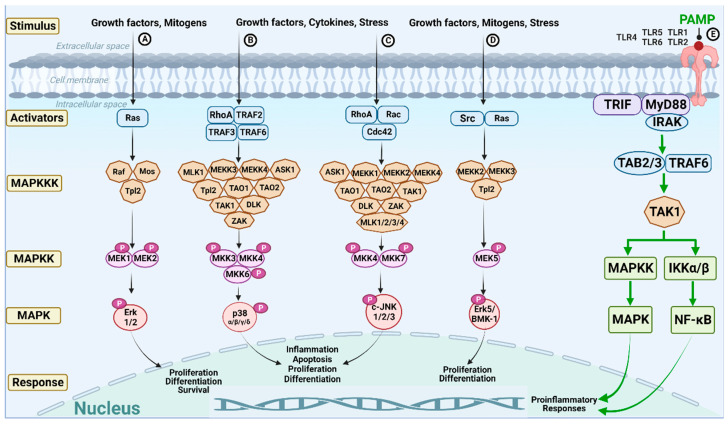
MAPK signaling pathways in mammalian cells: The MAPK signaling pathways are mediated by highly conserved MAPKs, which are activated sequentially. The figure depicts four canonical pathways organized in four tiers (**A**–**D**). Two non-canonical (Erk3/4 and Erk7/8) pathways are not shown. Extracellular signals, such as growth factors, cytokines, mitogens, and stress (e.g., UV, starvation, etc.), can stimulate the pathways by affecting plasma membrane receptors (not illustrated). Upon stimulation, the receptors transmit the signal to cytoplasmic proteins, termed ‘activators’ (e.g., the small GTP binding proteins belonging to the Rho and Ras families). These ‘activators’ then stimulate the downstream MAP kinase kinase kinases (MAPKKK), which then activate by phosphorylating (P) specific serine/threonine or tyrosine residues present in the activation loop of the downstream MAP kinase kinases (MAPKK), which then further activate the downstream MAP kinases (MAPK) by phosphorylating identical residues in their activation loop. It is worth pointing out that these pathways are interconnected. For instance, Tpl2 can signal the Erk1/2 and Erk5/BMK1 pathways. In addition, there are MAPK signaling pathways whose final targets are not nuclear, such as the myosin light chain kinase (MLCK) functioning downstream of Ras/MEK/Erk. Such pathways are not indicated in the figure, although they are affected by bacterial effectors, as in the case of Salmonella SteC. The activated MAPKs translocate into the nucleus, activating various transcriptional programs, which alter the gene expression, leading to diverse cellular responses, including cell growth, proliferation, differentiation, survival, apoptosis, and innate immune responses, such as proinflammatory responses against invading microbial pathogens. The activation of the MAPK and NF-ĸB signaling pathways by bacterial pathogen-associated molecular patterns (PAMPs) mediates the activation of the Toll-like receptors (TLR)/myeloid differentiation primary response 88 (Myd88)/TRIF (Toll/interleukin-1 receptor) signaling pathway is depicted (**E**). This pathway typically promotes the activation of transcriptional programs leading to proinflammatory responses. Ras—Rat sarcoma; RhoA—Ras homolog family member A; TRAF2/3/6—TNF receptor-associated factors 2/3/6; Cdc42—Cell division control protein 42; Rac1—Ras-related C3 botulinum toxin substrate 1; Src—Sarcoma; Raf—Rapidly Accelerated Fibrosarcoma; Mos—Moloney murine sarcoma virus; Tpl2—Tumor progression locus 2; MLK1/2/3/4—Mixed lineage kinase 1/2/3/4; MEKK1/2/3/4—MEK kinase 1/2/3/4; ASK1—Apoptosis signal-regulating kinase-1; TAO1/2—Thousand and one amino acid 1/2; DLK—Dual leucine zipper-bearing kinase; TAK1— transforming growth factor (TGF)—β-activating kinase-1; ZAK—Sterile alpha motif and leucine zipper containing kinase AZK; MEK1/2/5—MAPK and Erk kinase 1/2/5; MKK3/4/6/7—MAP kinase kinase 3/4/6/7; Erk1/2—Extracellular signal-regulated kinase 1/2; JNK—c-Jun N-terminal kinase; Erk5/BMK1—Extracellular signal-regulated kinase-5/ Big mitogen-activated protein kinase-1, PAMP—Pathogen-associated molecular patterns; TLR—Toll-like receptors; MyD88—Myeloid differentiation primary response protein 88; IRAK—Interleukin-1 receptor-associated kinase; TAB2/3—TGF-β activated kinase 1 (TAK1) binding protein 2/3; IKKα/β – Ikappa α and β kinases; NF-kB – Nuclear factor kappa-light-chain-enhancer of activated B cells.

**Figure 2 ijms-24-11905-f002:**
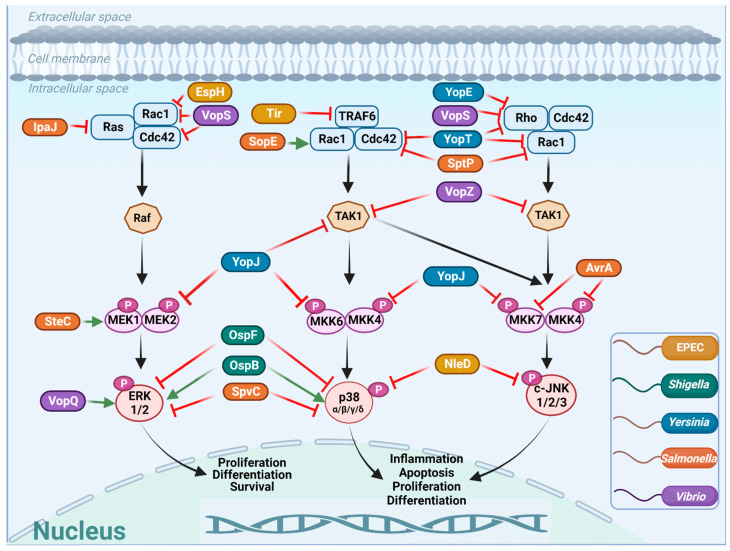
Modulation of mammalian MAPK pathways by effector proteins: This figure depicts the various components of MAPK pathways targeted by the bacterial effector proteins. The inhibitory (red Ʇ) or stimulatory (green →) effects of bacterial effectors on specific MAPK components are shown. The effector proteins are color-coded based on the bacterial colors (see inset). Notably, bacterial effectors seem to target multiple components of the canonical but not of the non-canonical pathways. Additionally, TAK1, a major MAPK component, appears to be targeted by various effectors. The reason for that could be the predominant involvement of TAK1 in eliciting proinflammatory responses. MAPKs can be indirectly affected by effectors whose mechanism of action on MAPKs is unclear. Examples are represented by the A/E pathogen effector Map and the *Cholerae* effector VopE, which regulate MAPK activity by targeting mitochondria. Additionally, some effector effects on MAPKs have been researched in yeast cells. For simplicity, both cases of MAPK targeting modes are not depicted in the Figure but are described in the text. For a similar reason, we did not include the effects of secreted toxins.

**Table 1 ijms-24-11905-t001:** Bacterial secreted effectors and toxins that modulate MAPK signaling.

Bacterium	Effector/ Toxin	Targeted Component of the MAPK Signaling Pathway (Activation/ Inhibition)	Enzymatic Activity	Mechanism	Host Response	Host Models	Ref.
** * Enteropathogenic Escherichia (E.) coli (EPEC) and enterohemorrhagic E. coli (EHEC) * **	**Tir**	TRAF6, Erk1/2, c-JNK, p38, NF-ĸB (**inhibition**)	Unknown	An ITIM motif in Tir interacts with SHP1&2, which inhibits the ubiquitination of TRAF6 and the downstream NF-κB/MAPK pathway	Inhibition of proinflammatory cytokine production	Mammalian cell culture, *C. rodentium*/murine	[38,42]
	**EspH**	Cdc42, Rac1, Erk1/2 (**inhibition**)	Rho-GAP through binding host Abr	Inhibits RhoGTPases and deactivates Erk1/2	Inhibition of innate immunity through inhibition of TNF-α- induced Erk signaling; inducing host cell cytotoxicity and death	Mammalian cell culture	[47,48]
	**Map**	Cdc42, MEK1, Erk1/2, p38 (**activation**)	Rho-GEF (WxxxE)	Activation of EGFR and MAPK signaling independent of Rho GTPase activation. MAPK is activated through mitochondrial cytotoxicity, the rise in cytoplasmic Ca^+2,^ and stimulation of ADAM10	Induction of apoptosis	Mammalian cell culture	[52]
	**NleH1/** **NleH2**	NleH1—Erk1/2, p38, NF-ĸB NleH2—p38, NF-ĸB (**inhibition**)	Kinase	The kinase activity deactivates NF-ĸB, but not the MAPKs; the mechanism of MAPK inhibition is not known	Improved recovery from colitis	Mammalian cell culture, murine	[53]
	**NleC**	p38, NF-ĸB (**inhibition**)	Zinc metalloprotease	Cleaves NF-ĸB; the mechanism of p38 inhibition is not known	Inhibition of IL-8 release	Mammalian cell culture, murine	[56,57,58,59]
	**NleD**	c-JNK, p38, NF-ĸB (**inhibition**)	Zinc metalloprotease	Cleaves the TXY motif in the activation loop of the MAPKs	Inhibition of AP-1-dependent gene transcription and innate immune responses	Mammalian cell culture	[60,61,62]
	**NleE**	TAK1, NF-ĸB (**inhibition**)	Cysteine methyltransferase	Inhibits TAB2/3 by methylation of a conserved cysteine in Npl4-like zinc finger domains of the complex, leading to the TAK1-mediated suppression of downstream NF-ĸB	Inhibition of IL-6, IL-8, and TNF production	Mammalian cell culture	[64,65,67,68]
** * Shigella flexneri * **	**OspF**	c-JNK, p38, Erk 1/2 (**inhibition**)	Phosphothreonine lyase	Removes a phosphorylated-threonine residue in the TXY motif of the activation loop in MAPKs	Inhibition of production of IL-8, c-Fos, CD44, and NF-ĸB1	Mammalian cell culture, yeast, guinea pig	[76,78,79,80,81]
	**OspB**	Erk1/2, p38 (**activation**)	Unknown	OspB activates Erk1/2 and p38, leading to the activation of inflammatory responses.	Promotes the production and secretion of metabolites involved in polymorphonuclear (PMN) leucocytes attraction	Mammalian cell culture, Guinea pig	[82]
	**IpaH9.8**	MAPKK Ste7 (**inhibition**)	E3 ubiquitin ligase	Promotes proteasome-dependent degradation of the MAPKK Ste7 in yeast and NF-ĸB and guanylate-binding proteins in mammalian cells	Suppression of MIP-2, IL-6, IL-1β	Mammalian cells, yeast, murine	[83,84,85,86,87]
** * Yersinia spp. * **	**YopJ/P**	MEK2, MKK4, MKK6, MKK7 and MAPKKK-TAK1, MKK Pbs2 (**inhibition**)	Acetyltransferase	Adds acetyl-coenzyme A to critical serine, threonine, and lysine residues in the activation loop	Induces programmed cell death and inhibits proinflammatory signaling through TNFα and IL-8	Mammalian cell culture, Drosophila, yeast	[96,97,98,99,100]
	**YopE**	Cdc42, RhoA, Rac1, c-JNK, Erk1/2 (**inhibition**)	Rho-GAP	Inactivates RhoGTPases by GAP activity, suppressing c-JNK and Erk1/2	Inhibition of the production of IL-8	Mammalian cell culture, murine	[101,102,103,104,106]
	**YopT**	Cdc42, RhoA, Rac1, c-JNK, Erk1/2 (**inhibition**)	Cysteine protease	Proteolytically cleaves the lipid modification of the RhoGTPases, resulting in their deactivation	Inhibition of the production of IL-8	Mammalian cell culture, murine	[99,104,105]
** * Salmonella enterica * ** ** serovar Typhimurium **	**SopE/** **SopE2** **and SopB**	Cdc42, Rac1, Erk1/2, p38, JNK (**activation**)	Rho-GEF	Activates RhoGTPases and downstream MAPKs to induce bacterial internalization and proinflammatory response	Production of IL-8	Mammalian cell culture	[118,119]
	**SteC**	MAPKK (**activation**)	Kinase	Phosphorylates MEK1 and MEK2, which activates the Erk/MLCK/Myosin IIB pathway	F-actin remodeling	Mammalian cell culture, murine	[120,121]
	**SptP**	Cdc42, Rac1, Raf-1, Erk1/2, c-JNK (**inhibition**)	Rho-GAP and tyrosine phosphatase	Inhibits the Raf-1/Erk1/2 pathway through C-terminal tyrosine phosphatase activity	Inhibition of IL-8 and TNF- α production	Mammalian cell culture	[122,123]
	**AvrA**	MKK4, MKK7, c-JNK (**inhibition**)	Acetyltransferase	Acetylates a specific serine residue of MKK4 and MKK7 and blocks their phosphorylation leading to the suppression of c-JNK and NF-ĸB	Inhibition of inflammatory responses and cell death	Mammalian cell culture, Drosophila, yeast, murine	[120,125,126]
	**IpaJ**	Ras (**inhibition**)	Unknown	Prevents the ubiquitination of Ras and phosphorylation of downstream MEK and Erk1/2	Downregulates proinflammatory responses, cellular growth, differentiation, cell survival, and apoptosis	Mammalian cell culture	[128]
	**SpvC**	Erk1/2, p38, c-JNK (**inhibition**)	Phosphothreonine lyase	Phosphothreonine lyase activity towards p38, Erk1/2 in vitro, and Erk1/2 only in vivo	Inhibition of proinflammatory cytokine production, neutrophil infiltration, and pyroptotic cell death	Mammalian cell culture, murine	[129,130,131,132,133]
** * Vibrio cholerae * **	**CTXB ***	c-JNK, p38 (**inhibition**)	Unknown	Induces the expression of MKP1 and inhibits the activation of c-JNK and p38	Inhibition of LPS-activated proinflammatory responses (TNFα and IL-6)	Mammalian cell culture	[139,140]
	**MARTX ***	Rho GTPases (**inhibition**)	Unknown	Inactivates Rho GTPases and downstream MAPK pathways	Inhibition of IL-8 production and intestinal inflammation	Mammalian cell culture	[141,142]
	**VopE**	Cell wall integrity-MAPK (CWI-MAPK) (**inhibition**)	Unknown	Disrupts the MAPK signaling pathway through an unknown mechanism	Unknown	Yeast	[144]
** * Vibrio parahaemolyticus * **	**VopQ**	Erk1/2 (**activation**)	Unknown	Activates the IRE1 branch of the unfolded protein response, resulting in the induction of Erk1/2	Unknown	Mammalian cell culture	[148]
	**VopS**	Rho GTPases (**inhibition**)	AMPylation	AMPylates Rho GTPases, resulting in the shutdown of the downstream MAPKs	Unknown	Mammalian cell culture	[148]
	**VopA**	Erk1/2, p38, c-JNK (**inhibition**)	Acetyltransferase	Acetylates a conserved lysine located in the catalytic loop of MAPKs	Suppress host innate immune responses, but not the NF-κB pathway	Mammalian cell culture, yeast	[150,151]
	**VopZ**	TAK1 (**inhibition**)	Unknown	Inactivates TAK1	Unknown	Mammalian cell culture	[152]
** * Listeria monocytogenes * **	**LLO ***	Raf, p38 (**activation**)	Unknown	Activates the Raf-MEK and p38 pathway	Activation of gene expression of NF-ĸB and the p38 pathway	Mammalian cell culture, murine	[159]

* in red rubricates represent the toxins.

## Data Availability

Not applicable.

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
