# Peer review of "Mitogen-Activated Protein Kinases (MAPKs) and Enteric Bacterial Pathogens: A Complex Interplay"

_ijms, 2023, doi:10.3390/ijms241511905_

Round 1
Reviewer 1 Report
Comments and Suggestions:
This manuscript explores the interplay between Mitogen-Activated Protein Kinases 1 (MAPKs) and Enteric Bacterial Pathogens. The review has been submitted with well-written content that effectively summarizes recent studies and presents fascinating findings.
However, there are some areas that require improvement. The authors should clearly state their statements and provide supporting original research references to strengthen the manuscript. Additionally, concise conclusions would enhance its overall impact.
Several references are incomplete and contain punctuation errors or fused words. It is crucial to rectify these issues and proofread the entire manuscript for grammatical errors. Moreover, the flow of sentences should be improved to enhance readability and ensure coherence with preceding sentences.
There are instances where the manuscript repeats information, which needs careful checking and revision.
Overall, this review manuscript offers valuable insights and suggests new directions for the development of anti-microbial treatments targeting MAPKs. It is an essential read for researchers working in the field of enteric bacterial pathogens, MAPKs, and their impact on bacteria and hosts.
Author Response
Comment 1: However, there are some areas that require improvement. The authors should clearly state their statements and provide supporting original research references to strengthen the manuscript. Additionally, concise conclusions would enhance its overall impact.
Reply: Original research papers have been indicated whenever statements concerning the involvement of bacterial elements (mainly toxins and effectors) have been studied to control MAPKs. Text indicating general information and insights not directly related to MAPKs was backed up with review article citations. This combination avoids excessive citations yet acknowledges studies contributing to advancing the topic. We added concise summaries of the effectors and their mechanism of action in each section (indicated in bold letters) and more general conclusions in the first two paragraphs of the ‘concluding remarks’. We hope that the combination of the two strategies enhances the clarity of the description of the complex biological systems and effects,
Comment 2. Several references are incomplete and contain punctuation errors or fused words. It is crucial to rectify these issues and proofread the entire manuscript for grammatical errors. Moreover, the flow of sentences should be improved to enhance readability and ensure coherence with preceding sentences.
Reply: We made changes in the text, and if the reviewer finds more errors, we would be delighted if he/she could identify them, and will make the required changes.
Comment 3. There are instances where the manuscript repeats information, which needs careful checking and revision.
Reply: We deleted the repeated information we could identify, and if the reviewer finds more, we would be grateful if he/she could indicate them specifically.
Reviewer 2 Report
The author summarized recent findings on enteric bacterial pathogens’ strategies to modulate MAPKs and host responses by raising many examples of bacterial infections. He/she mainly discussed on the actions of effectors and toxins secreted from various pathogens with a focus on the effects on MAPKs. He/she also discussed on the possibilities about application of these findings to treat various bacterial infection.
The contents are very interesting and would provide a huge impact on the research field of bacteria-host interaction. A number of interactions between different signaling pathways in enterocytes also seemed to be important to deeply understand the strategies of those pathogens. Thus, this review article seems worth to be accepted in this journal. However, the reviewer would recommend the author to complement about following points;
1. He/she did not describe what triggers the secretion of effectors and/toxins from bacteria. The mechanisms for starting of the secretion need to be described, though it has not been disclosed so much.
2. He/she did not describe how the effectors act on enterocytes, i.e., they can invade into cells, they can be received by individual receptors on cells etc.
3. He/she did not describe about genetic derivation of the effectors. He/she may also categorize the effector based on the chemical natures.
4. He/she did not describe about mechanisms for the attachment, colonization, distribution of bacteria on the surface of enterocytes. Are there specific receptors for individual bacteria on the cells? Whether responsible molecules on each side are known or not?
5. To the reviewer, it seems hard to simply understand roles of MAPKs in the bacterial infection or host defense, or both of them. What determines the positive/negative effects of MAPKs response? Does the quantity of the stress or response not affect final output?

Author Response
Comment 1. He/she did not describe what triggers the secretion of effectors and/toxins from bacteria. The mechanisms for starting of the secretion need to be described, though it has not been disclosed so much.
Reply. The mechanisms that trigger bacteria' secretion of effectors or toxins are complex and largely unknown. Since the interplay between these mechanisms and MAPK regulation is not known, these mechanisms would be beyond the scope for this article.
Comment 2. He/she did not describe how the effectors act on enterocytes, i.e., they can invade into cells, they can be received by individual receptors on cells etc.
Reply. Effector proteins are injected (translocated) from the bacterial cytoplasm into the host cells by secretion systems. Toxins are also secreted by secretion systems. A statement about that is now included in lines 72-76.
Comment 3. He/she did not describe about genetic derivation of the effectors. He/she may also categorize the effector based on the chemical natures.
Reply. The effectors are encoded by pathogenicity islands in the bacterial chromosome (lines 72-76), and they are proteins.
Comment 4. He/she did not describe about mechanisms for the attachment, colonization, distribution of bacteria on the surface of enterocytes. Are there specific receptors for individual bacteria on the cells? Whether responsible molecules on each side are known or not?
Reply. The mechanisms of bacterial attachment to the host cell are complex and only occasionally known. Nevertheless, they are likely not relevant to the activation of MAPK. Many parts of the manuscript indicate that external bacterial components trigger the MAPK pathways. For Shigella, we have added information concerning the bacterial external components (PGN) that trigger the MAPK/NFkB pathways by the NOD1/2 receptors (lines 201-203). In the case of Yersinia, the bacterial outer membrane protein InvA binds the host beta-1 integrin to facilitate Yersinia invasion and MAPK/NF-kB activation. A comment on that has been introduced in lines 256-261. For Listeria, the bacterial outer membrane InlB interacts with host cell c-Met and E-cadherin, and these interactions contribute to MAPK activation. A comment about that has been introduced in lines 433-435.
Comment 5. To the reviewer, it seems hard to simply understand roles of MAPKs in the bacterial infection or host defense, or both of them. What determines the positive/negative effects of MAPKs response? Does the quantity of the stress or response not affect final output?
Reply. Combined with the Figures and Table, the text comprehensively summarizes the modes by which the bacterial pathogens target MAPKs, and the consequence on the host. This includes the positive (e.g., the bacterial external components) and negative (e.g., contributed by enzymatic activities of the translocated effectors) effects on the MAPKs. The question regarding the relation between the quantity of stress or other cues and the extent of signaling output is excellent. It is well appreciated, for instance, that the MAPK/NF-kB sensing machineries amplify the signals, providing high system sensitivity for these signals. However, I guess that these systems are complex, and every cell type, under different conditions, will sense the cues and execute MAPK outputs differently. We did not find any data that quantified these events and therefore believe that this intriguing question remains open.